# Diagrammatic Reasoning for ALC Visualizations with Logic Graphs

## ABSTRACT

User studies show the demand for diagrammatic reasoning techniques for knowledge representation formats. OWL ontologies are highly relevant for Web 3.0, however, existing ontology visualization tools do not support diagrammatic reasoning, while existing diagrammatic reasoning systems utilize suboptimal visual languages. The purpose of this research is to facilitate the usage of OWL ontologies by providing a diagrammatic reasoning system over their visual representations. We focus on the ALC description logic, which covers most of the expressivity of the ontologies. As a visual language to reason about, we utilize Logic Graphs, which provide simplest visualizations regarding graph- and information-theoretic properties. We adapt the tableau algorithm to LGs to reason about concept satisfiability, prove the correctness of the proposed system and illustrate it with examples. The proposed diagrammatic reasoning system allows reasoning over ontologies, reducing complex concepts step by step, and identifying elements that produce a contradiction.

## CCS CONCEPTS

• **Information systems → Web Ontology Language (OWL)**; • **Human-centered computing → Graph drawings**; • **Computing methodologies → Description logics**.

## KEYWORDS

Ontology visualization, Diagrammatic reasoning, Existential graphs, Description logic, Tableau algorithm.

**ACM Reference Format:**
Anonymous Author(s). 2023. Diagrammatic Reasoning for ALC Visualizations with Logic Graphs. In *Proceedings of The Web Conference 2024 (WWW'24)*. ACM, New York, NY, USA, 9 pages. https://doi.org/XXXXXXX.XXXXXXX

## 1 INTRODUCTION

Due to their explainability and trustworthiness, knowledge-based systems are highly relevant for Web 3.0. They utilize machine-actionable knowledge representations, such as ontologies, for data exchange, validation and reasoning. However, to remain explainable and trustworthy, ontologies need to be also human-interpretable. That requirement refers to reasoning over ontologies as well.

*Example 1.1.* To illustrate the problem of human-readability for OWL reasoning, consider a concept of a person who is vegan, but not a vegetarian. Let us assume that a vegan is a person who eats only plants:

$$Vegan = Person \sqcap \forall eats.Plant,$$

and a vegetarian is a person who eats only plants or dairy

$$Vegetarian = Person \sqcap \forall eats.(Plant \sqcup Dairy).$$

Thus, a person who is vegan, but not a vegetarian, is defined by the following:

$$Person \sqcap \forall eats.Plant \sqcap \neg(Person \sqcap \forall eats.(Plant \sqcup Dairy)). \quad (1)$$

The result of logical inference for this concept with the Pellet [21] reasoner in the Protégé[1] interface is represented in Fig. 1. The interface shows that the concept is equivalent to the empty set, meaning it is inconsistent. There is also an explanation of this inference, see Fig. 2. But the explanation only repeats the original axiom in a slightly modified form.

One way to increase human-interpretability of ontologies is their visualization with diagrams. This approach may be applied to reasoning as well. A calculus for the diagrams can be defined that allows deriving new diagrams. A logical inference carried out on diagrams is called diagrammatic reasoning. The advantage of diagrammatic reasoning over symbolic inference is that the diagrams visually represent the semantics of the logical structures and, therefore, the inference is performed through transformations of the diagrams themselves. From a symbolic description, the information has to be calculated, while on a diagram, it can simply be seen. One can say that diagrammatic representations reflect logical relationships, while symbolic representations only describe them. One of the examples of a diagrammatic reasoning system is Ch. S. Pierce's existential graphs [22], based on first-order logic. The existing user studies prove the demand for diagrammatic reasoning techniques for other knowledge representation formats. The authors of [8] investigate how humans read constraint diagrams [10]. The results of the experiment show the need for augmentation of constraint diagrams in order to show in which order a diagram should be read. The experiments reported in [18] show that diagrammatic reasoning is useful and motivating for students in their attempts to learn the important notions of formal representation of knowledge.

The problem is that, on one hand, there are lots of tools for ontology visualization, such as VOWL [14], OntoGraf [7], Graphol [5, 13] or SOVA [11], they do not support diagrammatic reasoning. On another hand, there are diagrammatic reasoning systems for description logics [1], but their visual languages are in a way suboptimal.

The purpose of this research is to facilitate the usage of OWL ontologies by providing a diagrammatic reasoning system over their visual representations. We focus on the ALC description logic

---

[1]https://protege.stanford.edu/

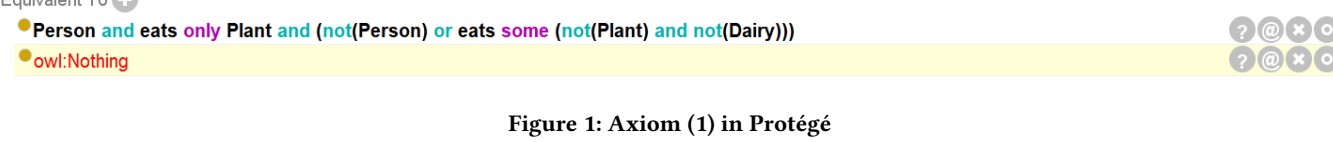

**Figure 1: Axiom (1) in Protégé**

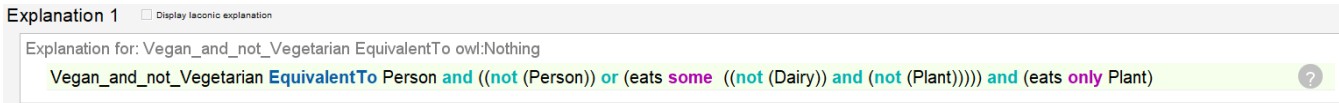

**Figure 2: Explanation of the inference in Protégé**

because it covers most of the expressivity of the ontologies. According to [28], 73,7% of the surveyed ontologies have the expressivity of ALC or lower. Particularly, we consider the concept satisfiability problem and thus, only TBox axioms. As a visual language to reason about, we utilize Logic Graphs (LGs) [15]. LGs is a semantic-oriented visual language for ontologies, complete with respect to the OWL DL language [16]. LGs adapt elements of existential graphs, graph theory, and Venn diagrams to represent the semantics of ontological structures. According to [3], visualizations in LGs are optimal with respect to their graph-theoretic properties such as the number of nodes, the number of edges and the depth of node enclosure.

Thus, the contribution of the present work is the following: i) developing a diagrammatic reasoning system for ontology visualization with LGs, based on the tableau algorithm of ALC, and ii) proving the correctness of this system.

## 2 RELATED WORK

### 2.1 Diagrammatic reasoning for FOL

Historically, the first diagrammatic reasoning systems were developed for first-order logic (FOL). There are such systems as Peirce's existential graphs (EGs)[22], conceptual graphs [23] and constraint diagrams [25]. Unfortunately, OWL ontologies have their logical foundation in DLs, which are less expressive than FOL. On the other hand, spider diagrams [27] correspond to monadic predicate logic, which is less expressive than DLs. Therefore, these diagrammatic reasoning systems do not perfectly fit OWL ontologies.

Nevertheless, to illustrate a diagrammatic reasoning approach, we consider the EGs reasoning system. Namely, we focus on its Alpha subsystem, which corresponds to propositional logic. In EGs Alpha, propositions are considered as nodes. A proposition can be asserted by writing it down on an area called a *sheet of assertion*. Two propositions on the sheet of assertion form a conjunction. Any proposition may be enclosed by a closed curve called a cut, which corresponds to a negation. For convenience, cut areas are shaded so that negations can be easily identified. A doubly cut area becomes unshaded.

All proofs in Peirce's system are based on formal rules by which one EG may be transformed into another preserving truth of propositions. There are three pairs of rules for EG Alpha. The rules about the insertion are numbered as 1i, 2i, 3i, and the inverse erasure rules are 1e, 2e, 3e:

- 1i. In a negative (shaded) area, one or more nodes may be inserted.
- 1e. In a positive (unshaded) area, one or more nodes may be erased.
- 2i. One or more nodes in any area may be iterated (copied) in the same area or into any area nested in it.
- 2e. Any node that could have been derived by rule 2i may be erased (whether or not a node had previously been derived by 2i is irrelevant).
- 3i. A double negation may be drawn around any collection of zero or more nodes in any area.
- 3e. Any double negation in any area may be erased.

This set of rules is proved to be sound and complete.

*Example 2.1.* We provide a simple example of proof. Consider a proposition "It rains, and if it rains, then it is cold". The corresponding graph is in Fig. 3a. In the first step, we can erase the inner instance of "it rains" using rule 2e, see Fig. 3b. The resulting graph contains a double cut, which according to rule 3e can be removed, see Fig 3c. Finally, we erase the proposition "it rains" with rule 1e and get the result in Fig 3d. Thus, the graph with the meaning "it rains, and if it rains, then it is cold" implies the graph with the meaning "it is cold".

### 2.2 Diagrammatic reasoning for ALC

There are works on diagrammatic reasoning for DLs. The authors of [6] investigate if spider and constraint diagrams as well as existential and conceptual graphs are compatible with DLs, and conclude that EGs are better suited for this purpose. They propose a diagrammatic calculus for a fragment of EGs corresponding to ALC, called Relation Graph (RG). There concepts are represented as labeled trees. Nodes of these trees correspond to concepts, unlabeled edges correspond to unions, and labeled edges to roles. As an RG is always a tree, it has more numerous and longer edges. Additionally, roles have to be represented with separate nodes in order to provide a possibility of their negation. This result in more complex representations.

Concept diagrams (CD) [17] are designed for expressing ontologies. They are based on constraint diagrams, but extended with variables and arrows. The authors of [4] provide a set of inference rules for concept diagrams, including rules for copying and deleting elements. They exemplify their system with case studies in [9]. In [24], the authors specify a system of concept diagrams for the

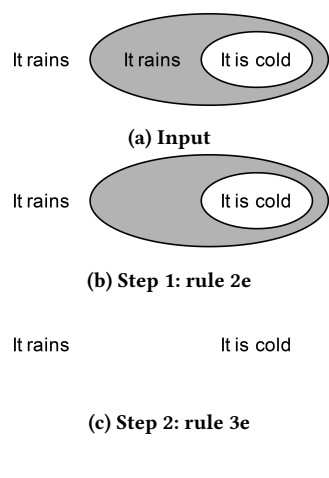

(a) Input

(b) Step 1: rule 2e

(c) Step 2: rule 3e

(d) Step 3: rule 1e

Figure 3: Example of a reasoning over EGs

OWL 2 language. In [19], it is demonstrated how to use concept diagrams for reasoning about inconsistencies in ontologies. In [26], a fragment of concept diagrams is specified for ALC. In [20], the authors presented an interactive theorem prover for concept diagrams, called iCon. The problem is that concept diagrams require separate nodes that represent a domain of discourse. Also they utilize overlapping of nodes, which significantly complicate the visualization.

Another visual language for ontologies is Logic Graphs (LGs) [15]. It adapts elements of existential graphs, graph theory, and Venn diagrams to represent the semantics of ontological structures. It is complete with respect to the OWL DL language [16]. Generating LGs for ontologies was implemented in [12].

*Example 2.2.* The Fig. 4, 5 and 6 represent the visualizations of the Example 1 in LGs, CD and RG respectively. As we do not consider graphical features such as forms of the nodes and spatial layout of graphs, we use a similar style for each of the visualizations.

We compare the visual languages for ALC with respect to their graph-theoretic properties. The results of the comparison are presented in Table 1. Each cell there contains a couple $(m, n)$, where $m$ is a number of nodes and $n$ is a number of edges required to represent a syntactic operation. We count a shading and an enclosure of one node into another as an edge. In case of CD, we take a lower bound, counting overlapping of two nodes also as an edge. However, the overlapped region could be considered as a separate node, subsequently increasing the number of edges. According to the comparison, LGs provide the simplest representation of ALC. More details on the evaluation of LGs can be found in [3].

## 3 BACKGROUND

### 3.1 ALC description logic

A short description of ALC DL [1] as a target syntax for visualization is provided. The vocabulary of ALC consists of countably infinite mutually disjoint sets of individuals $N_I$, roles $N_R$, and atomic concepts $N_C$. Concepts are recursively built using constructors $\neg$, $\sqcap$, $\sqcup$, $\forall$ and $\exists$.

A knowledge base $K = (T, A)$ consists of a TBox $T$ and an ABox $A$. A TBox is a finite set of general concept axioms of the form $C \sqsubseteq D$ or $C \equiv D$, where $C, D \in N_C$. An ABox is a finite set of concept assertions of the form $C(a)$, and role assertions of the form $R(a, b)$, where $a, b \in N_I$, and $R \in N_R$.

The syntax and semantics of ALC are summed up in Table 2. Here $I$ is an interpretation function and $\Delta$ is a domain.

Table 2: Syntax and Semantics of ALC

|  | Syntax | Semantic |
|---|---|---|
| Concept | $C$ | $C^I \subseteq \Delta^I$ |
| Role | $R$ | $R^I \subseteq \Delta^I \times \Delta^I$ |
| Complement | $\neg C$ | $\Delta^I \setminus C^I$ |
| Intersection | $C \sqcap D$ | $C^I \cap D^I$ |
| Union | $C \sqcup D$ | $C^I \cup D^I$ |
| Existential restriction | $\exists R.C$ | $\{a \in \Delta^I : \exists b(a, b) \in R^I \wedge b \in C^I\}$ |
| Universal restriction | $\forall R.\top$ | $\{a \in \Delta^I : \forall b(a, b) \in R^I \rightarrow b \in C^I\}$ |
| Concept inclusion | $C \sqsubseteq D$ | $C^I \subseteq D^I$ |
| Concept equivalence | $C \equiv D$ | $C^I \equiv D^I$ |
| Concept assertion | $C(a)$ | $a^I \in C^I$ |
| Role assertion | $R(a, b)$ | $(a, b)^I \in R^I$ |

### 3.2 Tableau algorithm for ALC

One of the most commonly used reasoning techniques in DLs that allows checking the satisfiability of concepts is the tableau algorithm [1]. The algorithm works by constructing a completion tree (CTree).

*Definition 3.1 (Completion tree).* A completion tree is a labeled tree $T = (V, E, L)$, where $(V, E)$ is a tree with a set of nodes $V$ and a set of edges $E$, and $L$ is a labeling function that assigns a label to all nodes and edges of $T$ as follows:

- $L(x)$ assigns a set of concept labels for a node $x \in V$,
- $L(x, y)$ assigns a role label for an edge $(x, y) \in E$.

A node $y \in V$ is called a successor of a node $x \in V$ in the tree $T$ if $(x, y) \in E$, and a node $y$ is called an $R$-successor of $x$ if $y$ is a successor of $x$ and $R \in L(x, y)$.

A CTree is initialized with a root node containing an initial concept **C** in negation normal form, i.e. $NNF(\mathbf{C})$. The following rules are iteratively applied in order to expand the CTree:

- $\sqcap$-rule: if $C_1 \sqcap C_2 \in L(x)$ then $L(x) = L(x) \cup \{C_1, C_2\}$,

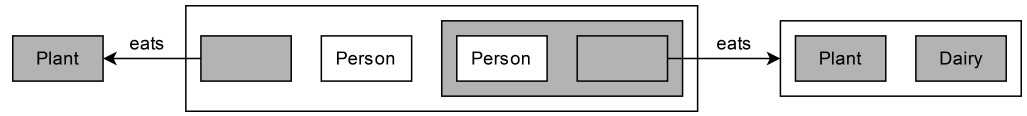

**Figure 4: Logic Graph for Example 1**

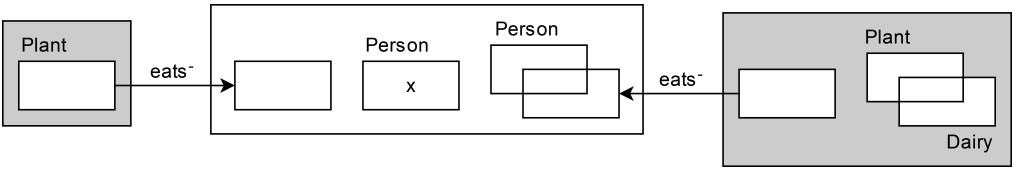

**Figure 5: Concept Diagram for Example 1**

**Table 1: Comparison of visual languages for ALC**

|     | Negation | Conjunction | Disjunction | Existential Restriction | Universal Restriction |
|-----|----------|-------------|-------------|-------------------------|-----------------------|
| LG  | *(1, 1)* | *(3, 2)*    | (3, 5)      | *(2, 1)*                | *(2, 3)*              |
| CD  | (2, 2)   | (3*, 3)     | *(3*, 4)*   | (4, 4)                  | (5, 6)                |
| RG  | (3, 1)   | *(3, 2)*    | (5, 4)      | (3, 2)                  | (5, 5)                |

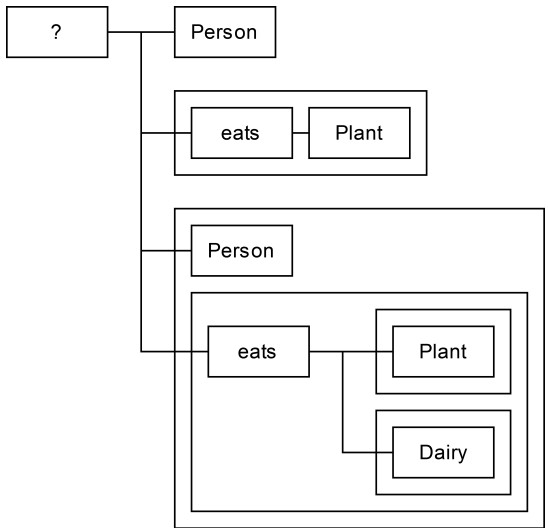

**Figure 6: Relation Graph for Example 1**

- ⊔-rule: if $C_1 \sqcup C_2 \in L(x)$ then $L(x_1) = L(x) \cup \{C_1\}, L(x_2) = L(x) \cup \{C_2\}$,
- ∃-rule: if $\exists R.C \in L(x)$ then create a new node $y$ with $L(x, y) = R$ and $L(y) = \{C\}$,
- ∀-rule: if $\forall R.C \in L(x)$ and there is an $R$-successor $y$ of $x$ then $L(y) = L(y) \cup \{C\}$.

The rules are applied until it is possible. If no rules can be applied, the tree is completed.

A concept **C** is satisfiable if the CTree is free of any local inconsistency called a clash.

*Definition 3.2 (**Clash**).* Given some concept $C$ and a node $x \in V$, a clash is a set $\{C, \neg C\} \in L(x)$.

If a CTree is complete and clash-free, the concept **C** is satisfiable, otherwise, **C** is inconsistent.

*Example 3.3.* Consider the axiom (1) about a vegan who is not a vegetarian. With the tableau algorithm, we can prove that this concept is inconsistent.

First, convert (1) in negation normal form:

$$Person \sqcap \forall eats.Plant \sqcap (\neg Person \sqcup \exists eats.(\neg Plant \sqcap \neg Dairy)).$$

The CTree for this concept is the following:

(1) Input: $L(x) = \{Person \sqcap \forall eats.Plant \sqcap (\neg Person \sqcup \exists eats.(\neg Plant \sqcap \neg Dairy))\}$.
(2) Apply ⊓-rule: $L(x) = L(x) \cup \{Person, \forall eats.Plant, \neg Person \sqcup \exists eats.(\neg Plant \sqcap \neg Dairy)\}$.
(3) Apply ⊔-rule:
  (a) $L(x_1) = L(x) \cup \{\neg Person\}$ – there is a clash between $Person$ and $\neg Person$;
  (b) $L(x_2) = L(x) \cup \{\exists eats.(\neg Plant \sqcap \neg Dairy)\}$.
(4) Apply ∃-rule to $x_2$: $eats(x_2, y)$ and $L(y) = \{\neg Plant \sqcap \neg Dairy\}$.
(5) Apply ∀-rule to $x$: $L(y) = L(y) \cup \{Plant\}$.
(6) Apply ⊓-rule: $L(y) = L(y) \cup \{\neg Plant, \neg Dairy\}$ – there is a clash between $Plant$ and $\neg Plant$.

Thus, no possible CTree for **C** is clash-free and, therefore, **C** is inconsistent.

### 3.3 Logic Graphs

The fragment of the LGs syntax corresponding to ALC is provided in Table 3. The space where a graph is located denotes the universe of objects. A rectangle denotes a concept, i.e. a set of objects. Shading denotes complement – objects that do not belong to the specified

set. Concepts can be nested within each other, meaning that objects of an outer rectangle belong to each inner rectangle. Note, that the space is closed under conjunction, thus, if there are $A$ and $B$ in a subspace, the whole subspace satisfies $A \sqcap B$. Arrows denote roles existentially restricted by default.

**Table 3: Logic graphs for ALC**

| ALC | Logic Graph |
|---|---|
| $C$ | C |
| $\neg C$ | C |
| $C1 \sqcap C2$ | C   D |
| $C1 \sqcup C2$ | C   D |
| $\exists R.C$ | —R→ C |
| $\forall R.C$ | —R→ C |

Complement, intersection, and existential restriction are sufficient for deriving other concept constructors. A representation of union is derived from representations of intersection and complement according to the tautology:

$$C_1 \sqcup C_2 \equiv \neg(\neg C_1 \sqcap \neg C_2),$$

and universal restriction is derived from existential one according to the tautology:

$$\forall R.C \equiv \neg\exists R.\neg C.$$

Thus, rectangles, shadings, and arrows in LGs are sufficient for representing ALC. For the LGs syntax for more expressive fragments of the OWL language see [16].

## 4 METHOD

In this section, a diagrammatic reasoning algorithm for LGs is presented. It allows checking satisfiability of ALC concepts in a visual and interactive way. The algorithm is based on the tableau algorithms for ALC. Analogously, the algorithm for LGs consists of constructing an LGTree, i.e. a tree where nodes are logic graphs.

*Definition 4.1 (**LGTree**).* A tree of logic graphs (LGTree) is a labeled tree $T = (V, E, L)$, where $(V, E)$ is a tree with a set of nodes $V$ and a set of edges $E$, and $L$ is a labeling function that assigns a label to nodes and edges of $T$ as follows:

- $L(x)$ assigns a set of logic graphs for a node $x \in V$,
- $L(x, y)$ assigns a role label for an edge $(x, y) \in E$.

An LGTree is initialized with a root node containing a logic graph $LG(C)$ for an initial concept $C$. We assume that the input concept includes all related background axioms, such as superclasses and their properties. This inclusion is possible due to the tautology:

$$C \sqsubseteq D \equiv \neg(C \sqcap \neg D).$$

As LGs have the graphic primitives only for complement, intersection, and existential restriction, the concept $C$ can not be used in $NNF$, but in the form that utilizes only complements, intersections, and existential restrictions, i.e. $\{\neg, \sqcap, \exists\}$-form, or $LG$-form.

There are rules for expanding the LGTree and thereby reducing the initial logic graph. Similarly to ALC, there is a rule for each constructor excluding complement, namely for intersection and existential restriction. However, as the $LG$-form is not a $NNF$, there are rules for their complements as well. Thus, there are two pairs of rules for reducing a logic graph. The rules for constructors themselves are denoted with '$+$', and the rules for their complements are denoted with '$-$'. For each rule, there are symbolic and visual representations. In the visual representations, outer rectangles denote separate logic graphs, and arrows of the form '$\Rightarrow$' between them represent the transformations of the graphs. For ease of perception, the elements to be removed are highlighted in green:

- $\sqcap^+$-rule – fig. 7. If there is an intersection node, remove it and place all its elements as separate nodes:

$$\{C \sqcap D\} \Rightarrow \{C, D\};$$

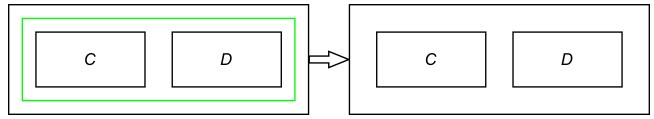

**Figure 7: $\sqcap^+$-rule**

- $\sqcap^-$-rule – fig. 8. If there is a node for a complement of intersection, remove it, and add a separate logic graph for each of its elements, placed as their complements:

$$\{\neg(C \sqcap D)\} \Rightarrow \{\neg C\}, \{\neg D\};$$

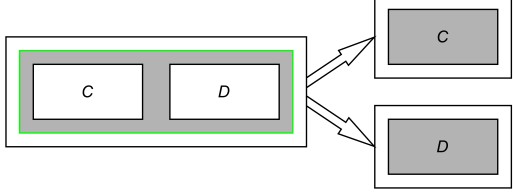

**Figure 8: $\sqcap^-$-rule**

- $\exists^+$- rule – fig. 9. If there is an edge for a role, add a new logic graph and place there the range-concept of the role:

$$\{\exists R.C\} \Rightarrow \{C\}_R;$$

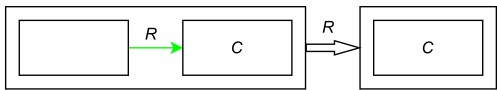

**Figure 9: $\exists^+$- rule**

- $\exists^-$-rule – fig. 10. If there is a complement for a domain of a role and there is an existing logic graph connected with this role, place a complement of the range-concept of the role in this existing logic graph:

$$\{\neg\exists R.C\}\{...\}_R \Rightarrow \{..., \neg C, ...\}_R.$$

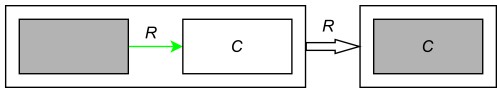

**Figure 10: $\exists^-$-rule**

The LGTree is complete, when it is not possible to apply any rule, i.e. when each logic graph is an atomic node or its complement. If the LGTree contains an LGClash, the initial concept **C** is inconsistent, otherwise, LGTree is clash-free, and **C** is satisfiable.

*Definition 4.2 (LGClash).* – Fig.11. There is an LGClash if a node $x$ of the LGTree contains a logic graph for a concept $C$ and a logic graph for its complement $\neg C$. For ease of perception, contradictory concepts are highlighted in red.

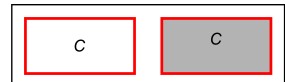

**Figure 11: LGClash**

## 5 VALIDATION

### 5.1 Correctness

For any deductive system, one of the fundamental questions is its correctness, i.e. its soundness and completeness. In this subsection, we prove the correctness of the proposed diagrammatic reasoning system regarding ALC by proving its equivalence to the tableau algorithm (TA), which is already proved to be correct:

THEOREM 5.1. $\vdash_{TA} G \Leftrightarrow \models_{ALC} G$,

where $G$ is an expression, $\vdash_{TA} G$ means that $G$ is deducible with TA, and $\models_{ALC} G$ means that $G$ is satisfiable in ALC. For proof, see [2].

LEMMA 5.2. $\vdash_{LG} G \Rightarrow \vdash_{TA} G$,

i.e. for each LG-inference there is an equivalent TA-inference.

PROOF. To prove that, we show that for each LG-rule there is a TA-inference with identical input and output:

- $\sqcap^+$-rule: $\{C \sqcap D\} \Rightarrow_{LG} \{C, D\}$.
  There is an equivalent $\sqcap$-rule in the ALC tableau algorithm:

$$\{C \sqcap D\} \Rightarrow_{ALC} \{C, D\}.$$

- $\sqcap^-$-rule: $\{\neg(C \sqcap D)\} \Rightarrow_{LG} \{\neg C\}, \{\neg D\}$.
  First, the formula $\neg(C \sqcap D)$ is put into negation normal form:

$$NNF(\neg(C \sqcap D)) = \neg C \sqcup \neg D.$$

  Then the $\sqcup$-rule is applied to $\neg C \sqcup \neg D$:

$$L(x) = \{\neg C \sqcup \neg D\} \Rightarrow_{ALC} L(x_1) = L(x) \cup \{\neg C\}, L(x_2) = L(x) \cup \{\neg D\}.$$

  Therefore,

$$\{\neg(C \sqcap D)\} \Rightarrow_{ALC} \{\neg C\}, \{\neg D\}.$$

- $\exists^+$-rule: $\{\exists R.C\} \Rightarrow_{LG} \{C\}_R$.
  There is an equivalent $\exists$-rule for ALC:

$$\{\exists R.C\} \Rightarrow_{ALC} \{C\}_R.$$

- $\exists^-$-rule: $\{\neg\exists R.C\}, \{...\}_R \Rightarrow_{LG} \{..., \neg C, ...\}_R$.
  Put $\neg\exists R.C$ into negation normal form:

$$NNF(\neg\exists R.C) = \forall R.\neg C.$$

  Apply the $\forall$-rule of ALC to $\forall R.\neg C$:

$$\forall R.\neg C \in L(x) \wedge R \in L(x, y) \Rightarrow_{ALC} L(y) = L(y) \cup \{\neg C\}$$

  Therefore,

$$\{\neg\exists R.C\}, \{...\}_R \Rightarrow_{ALC} \{..., \neg C, ...\}_R.$$

Thus, for each LG-rule, an equivalent inference is constructed using the tableau algorithm, therefore, for each LG-inference there is an equivalent TA-inference. □

LEMMA 5.3. $\vdash_{TA} G \Rightarrow \vdash_{LG} G$,

i.e. for each TA-inference, there is an equivalent LG-inference.

PROOF. Similarly, we show that for each TA-rule there is an LG-inference with identical input and output:

- $\sqcap$-rule: $\{C \sqcap D\} \Rightarrow_{ALC} \{C, D\}$.
  There is the equivalent $\sqcap^+$-rule for LGs:

$$\{C \sqcap D\} \Rightarrow_{LG} \{C, D\}.$$

- $\sqcup$-rule: $\{C \sqcup D\} \Rightarrow_{ALC} \{C\}, \{D\}$.
  First, $C \sqcup D$ in LG-form is $\neg(\neg C \sqcap \neg D)$, then applying $\sqcap^-$-rule:

$$\neg(\neg C \sqcap \neg D) \Rightarrow_{LG} \{C\}, \{D\}.$$

- $\exists$-rule: $\{\exists R.C\} \Rightarrow_{ALC} \{C\}_R$.
  There is the equivalent $\exists^+$-rule for LGs:

$$\{\exists R.C\} \Rightarrow_{LG} \{C\}_R.$$

- $\forall$-rule: $\{\forall R.C\}, ..._R \Rightarrow_{ALC} \{..., C, ...\}_R$.
  In LG-form $\forall R.C$ is $\neg\exists R.\neg C$. Applying $\exists^-$-rule to $\neg\exists R.\neg C$:

$$\neg\exists R.\neg C \Rightarrow_{LG} \{C\}_R.$$

Thus, for each LG-inference there is an equivalent TA-inference. □

LEMMA 5.4. $\vdash_{LG} G \Leftrightarrow \vdash_{TA} G$,

i.e. the diagrammatic reasoning algorithm for LGs is equivalent to TA (with respect to ALC).

PROOF. It comes from Lemma 5.2 and Lemma 5.3. □

THEOREM 5.5. $\vdash_{LG} G \Leftrightarrow \models_{ALC} G$,

where $\models_{ALC} G$ means that $G$ is satisfiable in ALC.

PROOF. It comes from Lemma 5.4 and Theorem 5.1. □

## 5.2 Examples

*Example 5.6.* To illustrate the diagrammatic reasoning for LGs, we refer to the same axiom (1). Though the interactive reasoning procedure can not be represented in the paper format, the corresponding LGTree is presented in Fig. 12. For convenience, coordinates in the form 'level.branch' and a rule currently applied are provided above each node. The tree has two branches, and each of them contains an LGClash, therefore, the initial logic graph is inconsistent.

*Example 5.7.* Consider another example to illustrate that the proposed algorithm does not mistakenly yield a contradiction if an initial logic graph is consistent. Thus, consider a person who is a vegan and a vegetarian:

$$Person \sqcap \forall eats.Plant \sqcap \forall eats.(Plant \sqcup Dairy). \qquad (2)$$

The corresponding LGTree is in Fig. 13. It also contains two branches, but both are clash-free, therefore, the concept (2) is satisfiable.

Thus, in contrast to the existing ontology visualization tools, the proposed diagrammatic reasoning system for LGs allows, first, reasoning over ontologies interactively, reducing complex concepts step by step, and second, identifying elements that produce a contradiction. Compared to other existing diagrammatic reasoning systems for ALC, the proposed one is based on the LGs visual language and the tableau algorithm.

## 6 DISCUSSION

**User experience.** LGs are an abstract syntax that can be implemented in various forms and layouts. Here, we aim at the functional feature of providing an interactive visual procedure for analyzing complex OWL axioms. For us, only the complexity of representations in graph- and information-theoretic terms is important. And in [3] it was demonstrated that LGs provide the simplest visualizations of OWL axioms. The readability of the visualizations is out of the scope of this research. Thus, a user study for evaluating the experience of using the proposed diagrammatic reasoning system would not be relevant. Particular representations, such as shapes and colors, could be altered in future to improve the user experience.

**Implementation.** Though the envisioned diagrammatic reasoning algorithm is not yet implemented, we suggest how it can be done. The paper [12] describes the generation of LGs for OWL ontologies. This approach produces intermediate DOT files that are further laid out. The diagrammatic reasoning algorithm proposed in the present research can be further implemented as operations over those intermediate DOT representations. Additionally, an LGTree can be priorly generated, laying out each possible transformation

of an LG on a separate web page. Then an interactive nature of the diagrammatic reasoning can be imitated with navigation through those pages. We provide an interactive demo[2] based on the Examples 5.6 and 5.7, which utilizes preliminary constructed LGs connected into an LGTree with hyperlinks.

**Complexity.** The tableau algorithm for ALC was proved to be PSPACE-complete [2]. Though TA differs from the diagrammatic reasoning algorithm for LGs regarding the input forms (TA requires NNF, while LG-inference requires LG-form, which includes complex negations), a similar analysis can be applied to the latter. Since each branch of an LGTree can be treated separately, the algorithm needs to store only one branch together with the direct successors of the nodes on this branch and the information on which of these successors must be investigated next. Since the number of branches and the depth of the LGTree are linear regarding the length of the input LG, the necessary information can be stored within polynomial space.

**Application scope.** By visualizing ALC, we cover the biggest part of the OWL usage. According to [28], 73,7% of the surveyed ontologies have the expressivity of ALC or lower. However, LGs syntax cover the whole OWL DL syntax [16] and in future work we plan to extend the proposed diagrammatic reasoning system for more expressive fragments of LGs. Additionally, we discuss only the concept satisfiability problem, thus ABox axioms are not considered here. However, the reasoning task related to ABox axioms such as classification or link prediction can be addressed in future work.

## 7 CONCLUSION

User studies show the demand for diagrammatic reasoning techniques for knowledge representation formats. OWL ontologies are highly relevant for Web 3.0, however, existing ontology visualization tools do not support diagrammatic reasoning, while existing diagrammatic reasoning systems utilize suboptimal visual languages.

The purpose of this research was to facilitate the usage of OWL ontologies by providing a diagrammatic reasoning system over their visual representations. We focused on the ALC description logic, which covers most of the expressivity of the ontologies. As a visual language to reason about, we utilized Logic Graphs, which provide simplest visualizations regarding graph- and information-theoretic properties. Further, we adapted the tableau algorithm to LGs to reason about concept satisfiability, proved the correctness of the proposed system and illustrated it with examples. In contrast to the existing ontology visualization tools, the proposed diagrammatic reasoning system allows reasoning over ontologies, reducing complex concepts step by step, and identifying elements that produce a contradiction. Compared to other existing diagrammatic reasoning systems for ALC, the proposed one is based on the LGs visual language and the tableau algorithm.

The future research will include i) designing a concrete syntax for LGs to improve user experience while using them and evaluating this syntax with user studies, ii) implementing the proposed reasoning system as an interactive visual reasoner for ontologies, iii) extending the diagrammatic reasoning system to the more expressive fragments of LGs and to ABox axioms.

---

[2]https://logic-graphs.github.io/diagrammatic-reasoning/

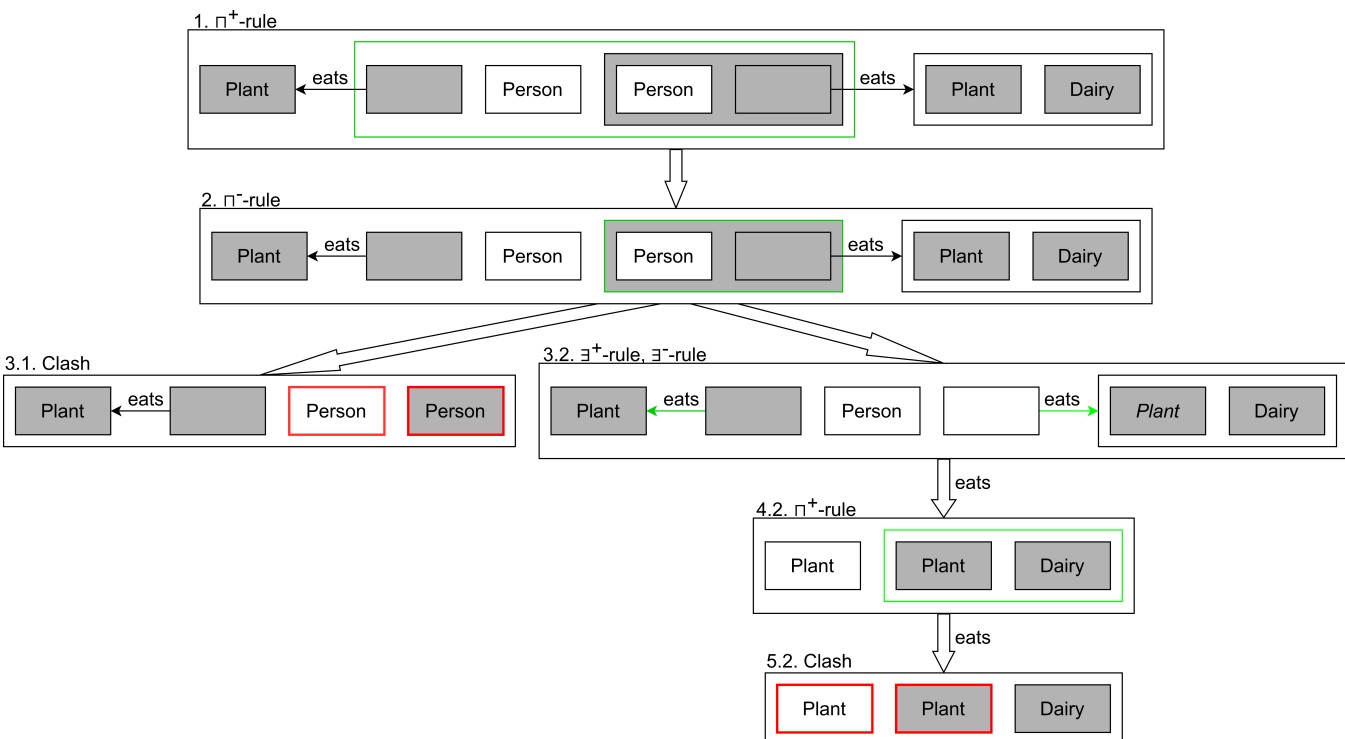

**Figure 12: LGTree for (1)**

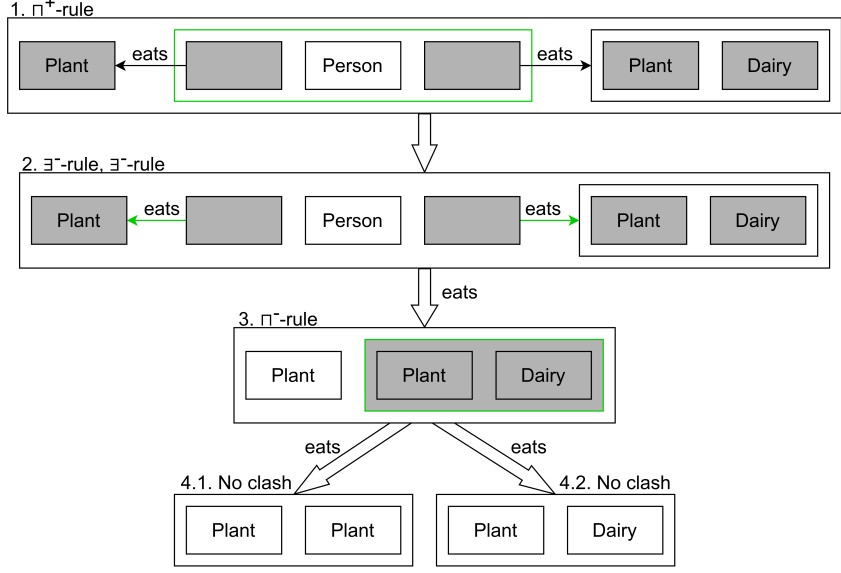

**Figure 13: LGTree for (2)**

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

Received 12 October 2023

