# OpenReview forum: "Diagrammatic Reasoning for ALC visualizations with Logic Graphs"
_ACM.org/TheWebConf/2024/Conference — TheWebConf24_

### Official Review · Reviewer_pLdk · 2023-11-22

**Novelty:** 4
**Technical Quality:** 5

**Review:**

The paper proposes a theoretical method to facilitate the usage of OWL ontologies by the help of a diagrammatic reasoning system
over the ontologies’ visual representations. The focus falls on the ALC description logic, as it covers the expressivity of most available OWL ontologies according to a survey.

The paper is overall well-written and not difficult to follow.

There’s an issue with the very first example in the introduction, because the concepts are ill defined. A vegan is a specific kind of vegetarian (meaning that a vegan IS indeed a vegetarian - you can actually check that) and if seen so, your reasoning will not yield the empty set. Hence, I’m struggling to grasp the motivation and application of the proposed algorithm.

The related work section contains formal parts which are not suited for this section — consider moving them to the Background section and focus here only on related approaches within the scope of the paper and the position of the latter wrt the former.

The paper only includes a formal evaluation of the proposed approach, no experiments are given, nor at least a set of clearly defined use-cases that could help understand the practical value of the approach. Indeed, as the authors mention, the algorithm is not yet implemented, which makes less convincing its applicability.

**Questions:**

- What is the practical value of the proposed method?
- How does this system compare to NLP-based approaches (pretrainded large language models) who could show similar functionalities on an arguably lesser cost?
- What is the envisioned experimental evaluation setting?

**Ethics Review Description:**

no issue

**Reviewer Confidence:**

3: The reviewer is confident but not certain that the evaluation is correct

**Scope:**

3: The work is somewhat relevant to the Web and to the track, and is of narrow interest to a sub-community

---

### Official Review · Reviewer_rhae · 2023-11-23

**Novelty:** 4
**Technical Quality:** 7

**Review:**

The paper proposes a diagrammatic reasoning method for ALC, an expressive description logic, inspired by the tableau algorithm, and prove that it is sound and complete. The method is based on the Logic Graphs, a visual language for ontologies that derives from Peirce's existential graphs, with some additions.
The paper is very well written, easy to follow, and self-contained. The contribution is clearly positioned with respect to the relevant literature. The paper is technically sound: I have checked all the proofs and they are correct.
Having said that, I find that the diagrammatic explanation of proofs, as proposed in the paper, are intuitive for a user provided that the user is familiar with the tableau algorithms of description logics, or, better, with the four rules of the Logic-Graphs-based diagrammatic reasoning system. It remains to be debated to what degree one can expect this from a casual user, while it is not obvious that a description logic expert would really benefit from a visual explanation of inferences...

**Questions:**

What would be the target user of the proposed reasoning system?

**Reviewer Confidence:**

4: The reviewer is certain that the evaluation is correct and very familiar with the relevant literature

**Scope:**

3: The work is somewhat relevant to the Web and to the track, and is of narrow interest to a sub-community

---

### Official Review · Reviewer_2Pjn · 2023-11-23

**Novelty:** 4
**Technical Quality:** 5

**Review:**

Summary:

This paper focuses on enhancing the usage of OWL ontologies through a diagrammatic reasoning system. This system is designed to make the ontologies more human-interpretable, which is crucial for knowledge-based systems, especially in the context of Web 3.0. The research specifically addresses ALC (Attributive Concept Description Language with Complements) description logic, which is a fundamental part of OWL ontologies.

The main contribution of the research includes:

1. Developing a diagrammatic reasoning system for visualizing ontologies using Logic Graphs (LGs).
2. Adapting the tableau algorithm for ALC to these Logic Graphs.
3. Proving the correctness of this system

Strengths:

1. The diagrammatic reasoning system improves the human interpretability of OWL ontologies, which is important for trustworthiness and explainability in Web 3.0 contexts.
2. The use of LGs provides simpler visualizations for reasoning about ontologies, making complex concepts more accessible.
3. The adaptation of the tableau algorithm for ALC to LGs offers a practical tool for reasoning about concept satisfiability in ontologies.

Weaknesses:

1. As I am not well-versed in logical proofs, ALC, and OWL domains, I am unable to fully assess the correctness of the algorithmic proof process proposed by the authors.
2. I think that the logic tree-based reasoning algorithm proposed by the authors is significantly innovative. It offers powerful interpretability for knowledge reasoning, especially fitting the needs of the current learning and artificial intelligence fields. This methodology might provide a new perspective for many logic reasoning tasks.
3. The discussion section mentions issues related to the implementation of the algorithm. I would like the authors to elaborate further on the specific application plans of this algorithm in practice, as well as how it integrates with existing technologies and tools.
4. I am particularly interested in how this work could offer new insights or potential solutions to the reasoning challenges of current Large Language Models (LLM). Could the authors discuss this aspect, envisioning the potential contributions and scope of this algorithm in future LLM research?

**Questions:**

see weakness.

**Reviewer Confidence:**

3: The reviewer is confident but not certain that the evaluation is correct

**Scope:**

3: The work is somewhat relevant to the Web and to the track, and is of narrow interest to a sub-community

---

### Official Review · Reviewer_ozdX · 2023-11-24

**Novelty:** 6
**Technical Quality:** 7

**Review:**

Pros: very clear paper, very well written, provides examples and application scenarios
Cons: no obvious shortcomings. Authors have openly stated about the status of their work, and outlined scenarios for the future work

In summary I see that this paper is publishable as it is. The topic should be of great interest for the Web Conference participants, especially since it provides clear and well communicated application scenarios and examples in addition to (outstanding) presentation about their diagrammatic reasoning algorithm.

**Questions:**

Some draft visualizations would be interesting to see already in this paper, or at least in the very near future in your next papers.

**Ethics Review Description:**

-

**Reviewer Confidence:**

4: The reviewer is certain that the evaluation is correct and very familiar with the relevant literature

**Scope:**

3: The work is somewhat relevant to the Web and to the track, and is of narrow interest to a sub-community

---

### Official Review · Reviewer_6QzH · 2023-11-27

**Novelty:** 3
**Technical Quality:** 4

**Review:**

The paper proposes a new description annotation language for ACL description logic.
The paper provides some details about the proposed new annotation language. However, it does not provide any sound reason why this new language might be needed or useful. It describes some of the other description language but it does not provide any evidence as to why those visual languages are suboptimal.

**Questions:**

How can the authors know that the proposed new language can increase human understandability of the underlying logic and its expressivity?

**Reviewer Confidence:**

3: The reviewer is confident but not certain that the evaluation is correct

**Scope:**

3: The work is somewhat relevant to the Web and to the track, and is of narrow interest to a sub-community

---

### Decision · Program_Chairs · 2024-01-22

**Decision:**

Accept

**Comment:**

The paper proposes a new diagrammatic reasoning system for ALC description logic, which is a core part of OWL ontologies. The system is based on Logic Graphs, a visual language that aims to improve the human interpretability and accessibility of ontologies. The paper provides a formal proof of the soundness and completeness of the system, as well as some examples and application scenarios.

 The paper is written clearly, has high technical quality and has a nice contribution to the field. There are minor improvements suggested, such as providing some draft visualizations or discussing the target users of the system. I believe these additions will make the paper better, and I strongly recommend the authors to add them in the final version.

 Based on this summary, I would recommend accepting this paper.